# Therapeutic Screening of Herbal Remedies for the Management of Diabetes

**DOI:** 10.3390/molecules26226836

**Published:** 2021-11-12

**Authors:** Mahmoud Balbaa, Marwa El-Zeftawy, Shaymaa A. Abdulmalek

**Affiliations:** 1Department of Biochemistry, Faculty of Science, Alexandria University, Alexandria 21511, Egypt; shimaa_salamy@yahoo.com; 2Biochemistry Department, Faculty of Veterinary Medicine, New Valley University, New Valley 72511, Egypt; marwa@vet.nvu.edu.eg; 3Center of Excellency for Preclinical Study (CE-PCS), Pharmaceutical and Fermentation Industries Development Centre, The City of Scientific Research and Technological Applications, Alexandria 21511, Egypt

**Keywords:** diabetes mellitus, *Nigella sativa*, curcumin, antidiabetic, hypoglycemic, insulin signal pathway

## Abstract

The study of diabetes mellitus (DM) patterns illustrates increasingly important facts. Most importantly, they include oxidative stress, inflammation, and cellular death. Up to now, there is a shortage of drug therapies for DM, and the discovery and the development of novel therapeutics for this disease are crucial. Medicinal plants are being used more and more as an alternative and natural cure for the disease. Consequently, the objective of this review was to examine the latest results on the effectiveness and protection of natural plants in the management of DM as adjuvant drugs for diabetes and its complex concomitant diseases.

## 1. Introduction

Diabetes mellitus (DM) is a chronic endocrine disorder, and its adverse effects currently occupy a huge challenge for prevention and/or treatment [1]. It is categorized into three main forms, type I, type II, and recently type III DM [2]. The acquired form of DM is type II, as it chiefly results from insulin resistance syndrome [3]. The various fatal adverse effects of type II DM include diabetic foot [4], diabetic bone disease [5], diabetic neuropathy [6], and declined resistance to bacterial and viral infection via affecting the innate immunity [7]. The molecular mechanistic pathway of type II DM is attributed mainly to defects in the kinase molecular signaling pathways, PI3K, p38MAPK, PKA, and calmodulin kinase, which influence glucose metabolism and insulin action [8].

There are multiple synthetic antidiabetic therapeutic families such as sodium–glucose co-transporter-2 inhibitors [9], dipeptidyl peptidase-4 inhibitors [10], glucagon-like peptide 1 analogs [11], sulfonylureas [12], thiazolidinediones [13], and biguanides [14]. However, for nearly 20 years, the new science of natural therapy has been highlighted to minimize some chronic diseases including type II DM [15]. The active ingredients in natural therapy may have antidiabetic activity, e.g., nonflavonoid polyphenols such as curcumin, tannins, lignans, and resveratrol [16] or flavonoids such as anthocyanins, epigallocatechin gallate, quercetin, naringin, rutin, and kaempferol [17].

In brief, most polyphenols and flavonoids exhibit their antidiabetic influence via improving the glucose control and insulin sensitivity [16], inhibiting oxidative stress [17], reducing inflammatory cytokine levels [18], inhibiting α-amylase and α-glucosidase activity [19], and increasing tyrosine phosphorylation of insulin receptor (IR) [20].

## 2. Diabetes Mellitus

### 2.1. Prevalence, Types, Symptoms, Pathophysiology, and Molecular Mechanism of DM

DM has received huge interest from most scientists as it is an international major public health threat. It is termed the silent killer, and it is predicted that, by the year 2030, the number of diabetic individuals will reach 578 million (700 million in 2045) [21]. DM is a chronic metabolic disorder, and there are two identified types, insulin-dependent DM (IDDM) and noninsulin-dependent DM (NIDDM). IDDM is an autoimmune disease caused as a result of the destruction of β-cells of the islets of *Langerhans* in the pancreas [22]. On the other hand, NIDDM occurs due to stress factors, obesity, and hormonal imbalance in which there is an overproduction of both insulin and amylin hormones from β-cells of the islets of *Langerhans* [23,24], as well as a reduction in adiponectin, calcium (Ca^2+^), and 25-hydroxyl vitamin D [25]. In recent years, Alzheimer’s disease was designated type III DM [26], and it is usually marked by amyloid-β plaques and phosphorylated-tau protein accumulation in the hippocampus of the brain [27]. Other types of DM may be temporary, such as gestational DM, which occurs in the second or third trimester of pregnancy in females and typically disappears after parturition [28]. Furthermore, in some conditions, DM results from complete or partial dissection of the pancreas as a consequence of some diseases related to the pancreas, such as tumors or severe inflammation [29].

In summary, the inability of pancreatic β-cells to produce insulin in IDDM [30] or insulin resistance [31] is implicated in the failure of insulin to perform its function, leading to hyperglycemia, polyuria, weight loss or increase, polydipsia, delayed wound healing, and blurred vision [32]. Hyperglycemia itself leads to an increase in the production of advanced glycation end-products (AGEs) and their receptors [33]. In this regard, especially in NIDDM, this is accompanied by the promotion of free radicals in the mitochondrial matrix that damage multiple biomolecules of the cell such as deoxyribonucleic acid (DNA), lipids, and proteins [34]. Consequently, this increases susceptibility to chronic inflammation and apoptosis, as well as impairs the function of various body organs [35].

On the other hand, AGEs and their receptors increase the activity of nicotinamide adenine dinucleotide phosphate (NADPH) oxidases and their messenger ribonucleic acid (mRNA), as well as arachidonic acid pathways [36]. The interaction of AGEs with the receptors of advanced glycation end-products (RAGEs) leads to the stimulation of some cell signal transduction pathways such as protein kinase C (PKC), phosphatidylinositol 3-kinase/protein kinase B (PI3K/Akt) [37], p38 mitogen-activated protein kinase (p38 MAPK) [38], extracellular signal-related kinases (ERK) [39], RhoA/Rho-kinase which activates many downstream kinases and mediates Ca^2+^ sensitization [40], Janus kinase/signal transducer and activator of transcription (JAK/STAT), and suppressor of the cytokine signaling 3 (SOCS3) [41]. Furthermore, there is dysregulation of 5′-adenosine monophosphate-activated protein kinase (5′-AMPK) activity via inhibition of gluconeogenesis genes [42], downregulation of glucose transporter-4 (GLUT-4) [43], stimulation of lipogenesis through elevation of HMG CoA reductase activity [44], and initiation of mitochondrial axonal cell death [45]. The consequence of these activated signals are (i) an increase in the level of transcription factors including nuclear factor-κB (NF-κB) [46] and early growth response-1 (Egr-1) protein, which is a vital zinc finger transcription factor [47], (ii) alteration of cell metabolism, and (iii) induction of inflammation, apoptosis, and proliferation by the NOD-like receptor protein-3 (NLRP-3) inflammasome [48]. Tumor necrosis factor-alpha (TNF-α), monocyte chemoattractant protein-1 (MCP-1), interleukin-6 (IL-6), and interleukin-1 beta (IL-1β) are among the cytokines produced [49]. These cytokines impair insulin signaling and peripheral glucose uptake and contribute to insulin resistance, lipolysis, and hepatic glucose production [50].

Moreover, hyperglycemia in NIDDM is a hazardous issue that disturbs the genetic expression responsible for insulin secretion, e.g., sirtuin-1 (Sirt-1) and glucose transporter-2 (GLUT-2), in β-cells [51]. It also activates the signaling pathway of insulin in adipose tissue and skeletal muscle, e.g., glucose transporter-4 (GLUT-4), which carries glucose from the cytoplasm to the membrane, and peroxisome proliferator-activated gamma receptor (PPAR-γ) [52], or in hepatic tissue, e.g., insulin receptor substrate-1 (IRS-1) serine/threonine/Akt-1 and phosphoenolpyruvate carboxykinase (PEPCK) [53]. The molecular mechanism of insulin resistance is illustrated in Figure 1.

### 2.2. Complications of DM

Untreated DM harms the minute blood vessels of some organs such as the kidney, heart, eye, and nervous system [54]. Hence, diabetic nephropathy [55], cardiomyopathy [56], retinopathy [57], and diabetic foot infection [58] are well-known adverse outcomes. Furthermore, vagus nerve atrophy may occur as an outcome of neuronal, autoimmune damage, and oxidative stress [59]. DM is connected to various musculoskeletal ailments, such as joint stiffness, gouty arthritis, osteoarthritis, rheumatoid arthritis, and diabetic amyotrophy [60]. In some cases, the negative effects of DM have been linked to the gut, where there has been a decrease in butyrate-producing bacteria and an increase in opportunistic pathogens [61]. moreover, the incidence of cancer may be a consequence of DM in some late stages [62]. A decrease in the salivary flow and elements is also obvious in diabetic individuals [63]. Furthermore, diabetic ketoacidosis and hyperglycemic hyperosmolar syndrome are both considered dangerous DM risks due to insulin deficiency, which results in the formation of ketone bodies and the occurrence of metabolic acidosis [64]. In some diabetic individuals, low immunity is also documented to make them more vulnerable to invasive fungal infections, such as the filamentous fungus *Syncephalastrum racemosum*, that affect the gastrointestinal tract [65].

Moreover, in 2020, it was confirmed that diabetic patients are very likely to be infected with COVID-19 due to their immune impairment [66]. Normally, the angiotensin-2 conversion enzyme (ACE-2) is expressed in β-pancreatic cells, and the SARS-Cov-2 virus binds primarily to ACE-2, causing damage to β-pancreatic cells [67]. It should be noted that, by stimulating oxidation free radicals and hypoxia-inducible factor 1 alpha (HIF-1α), DM enhances the replication of the virus [68]. Certain NIDDM conditions in different tissues are shown in Figure 2 and Figure 3.

## 3. Natural Therapy: A Safe Tool for DM Management

Today, due to their improved cost-effectiveness and avoidance of side-effects of certain drugs, medicinal plants may be used in the handling of DM. As shown in Figure 4, some herbal plants were found to improve hyperglycemia and insulin resistance via the AMPK signaling pathway.

### 3.1. Nigella Sativa (NS)

NS is often known as black cumin, belonging to the Ranunculaceae family, which grows extensively in many countries; it has many traditional uses as a spice and food preservative [69]. NS seeds have many biological functions, including carminative, stimulant, analgesic, antipyretic, and diuretic functions [70]. A complex blend of fatty acids, vitamins, pigments, and volatile components constitutes NS oil (NSO), which includes thymoquinone (TQ) and its associated compounds, thymol and dithymoquinone. It is important for the treatment of many diseases such as tumors, gastrointestinal disorders, cirrhosis, hepatitis, and chemical poisoning [71]. NSO also exhibited in vivo antidiabetic and neuroprotective effects in an animal model of experimental diabetes [72,73]. NS seed extract also had a beneficial effect on the liver [74]. NS regenerates β-cells of the pancreas during hyperglycemia as a consequence of its high polyphenol content, which enhances the metabolic process of carbohydrates and lipids [75] and its ability to hinder the upregulation of gluconeogenesis enzymes [76].

Several processes involving NSO itself or its main active ingredient, TQ, are responsible for the antidiabetic activity of NSO. Via stimulation of AMPK phosphorylation in hepatic and muscle tissues, NSO can increase insulin sensitivity [77]. Furthermore, NSO improves GLUT-4, insulin-like growth factor-1, and phosphatidyl inositol-3-kinase (PI3K) [78]. By inhibiting sodium–glucose co-transporters, NSO decreases glucose absorption from the intestine [79]. Another theory clarified that the decrease in the amount of glucose by NSO is due to its inhibitory effect on α-glucosidase [80]. NSO increases PARP-γ in the adipocyte and inhibits an enzyme that degrades insulin considered a cause of hyperglycemia [81]. Because of its unsaturated fatty-acid content and the downregulation of the 3-hydroxy-3-methylglutaryl-coenzyme reductase gene, which inhibits cholesterol oxidation and triacylglycerol lipoproteins, NSO affects hyperlipidemia caused by DM [82].

The oxidative stress present in DM is due to substantial production of the reduced form of nicotinamide adenine dinucleotide (NADH) that disrupts the equilibrium between NADH and its oxidized form NAD^+^, thus resulting in oxidative stress. Therefore, it is a redox imbalance disease [83]. Via the NADP-dependent redox cycle, TQ in NSO can re-oxidize NADH and, thus, decrease the NADH:NAD^+^ ratio. The re-oxidation of NADH to NAD^+^ by TQ stimulates glucose and fatty-acid oxidation, as well as Sirt-1-dependent pathways [84]. Moreover, NAD^+^ activates Sirt-1, which is an NAD^+^-dependent histone deacetylase that plays a key role in controlling both carbohydrate and lipid metabolism, as well as the secretion of adiponectin and insulin, and that protects pancreatic β-cells from oxidative stress and inflammation by inhibiting NF-κB activity [85]. The anti-inflammatory effect of NS during DM is notably linked with its repressing influences on cyclooxygenase and 5-lipoxygenase pathways, reducing nitric oxide, MCP-1, and TNF-α production and inhibiting IL-1β and IL-6 [86]. Furthermore, NS disrupts some DM complications such as nephropathy through upregulation of vascular endothelial growth factor-A (VEGFA) and transforming growth factor-β (TGF-β1) [87]. The molecular mechanistic pathways of the antidiabetic effect of NS are reported in Figure 5.

### 3.2. Berberine (BER)

BER is a quaternary ammonium isoquinoline alkaloid, which is present in some plant families such as Berberidaceae, Papaveraceae, Ranunculaceae, Rutaceae, and Menispermaceae [88]. BER achieves notable effects in treating and/or preventing various metabolic factors such as DM, hyperlipidemia, obesity, liver dysfunction, and some diseases related to disorders in nucleic acid metabolism [89]. In this review, we focus on the antidiabetic targets of BER that have multiple pathways. BER promotes insulin secretion, glucose uptake, and glycolysis [90], and it can also improve glycogenesis as a consequence of the inactivation of glycogen synthase kinase enzyme [91]. On the other hand, it prevents gluconeogenesis due to the reduction in its key regulatory enzymes, glucose-6-phosphate dehydrogenase and PEPCK [92]. Furthermore, BER reduces insulin resistance by upregulating PKC-dependent IR expression [93]; by blocking mitochondrial respiratory complex I, the adenosine monophosphate/adenosine triphosphate (AMP/ATP) ratio increases, thereby stimulating AMPK [94]. Hence, activated AMPK regulates transcription of uncoupling protein 1 in white and brown adipose tissue [95] and helps the phosphorylation of acetyl-CoA carboxylase (ACC) and carnitine palmitoyltransferase I enzymes, causing a reduction in lipogenesis and an increase in fatty-acid oxidation [96]. Via retinol-binding protein-4 and phosphatase and tension homolog downregulation, as well as sirt-1 activation, BER has a hypoglycemic function, thus improving insulin resistance in skeletal muscles [97].

Another mechanism of BER antidiabetic influence is attributed to its ability to regulate both short-chain fatty acids and branched-chain amino acids [98], whereby it diminishes the butyric acid-producing bacteria that destroy the polysaccharides [99]. A previous study displayed the role of BER in preventing cholesterol absorption from the intestine through improving cholesterol-7α-hydroxylase and sterol 27-hydroxylase gene expression [100]. Moreover, BER provides a vigorous defense against insulin resistance via the normalization of protein tyrosine phosphatase 1-B [101] and PPAR-γ/coactivator-1α signaling pathways that enhance fatty-acid oxidation [102]. Additionally, it was illustrated that BER adjusts GLUT-4 translocation via AS160 phosphorylation as a consequence of AMPK activation in insulin-resistant cells [103].

During DM there is a relationship between inflammation and oxidative stress which leads to the creation of proinflammatory cytokines such as IL-6 and TNF-α [104]. It was reported that BER counteracts some inflammatory processes where it attenuates NADPH oxidase (NOX) that is responsible for reactive oxygen species (ROS) generation, thereby decreasing AGEs and increasing endothelial function in DM [105]. BER displayed a tendency to ameliorate the inflammation resulting from DM via various pathways, e.g., suppression of phosphorylated Toll-like receptor (TLR) and IkB kinase-β (IKK-β) that is responsible for NF-κB activation; thus, BER interferes with the serine phosphorylation of IRS and diminishes insulin resistance [106]. Moreover, BER activates P38 that inhibits nuclear factor erythroid-2 related factor-2 (Nrf-2) and heme oxygenase-1 (HO-1) enzyme blockage, leading to proinflammatory cytokine production [107]. In addition, BER inhibits activator protein-1 (AP-1) and, thus, suppresses the production of cyclooxygenase-2 (COX-2) and MCP1 [108]. It was stated that BER alleviates some DM complications due to its capability of attenuating DNA necrosis in different affected tissues and enhancing the cell viability [109]. It was shown that BER protects the lens in diabetic eyes from cataract incidence by improving the polyol pathway through inactivation of the aldose reductase enzyme responsible for the conversion of glucose into sorbitol that degenerates the lens fiber [110]. Figure 6 shows some of the hypoglycemic mechanisms of BER mentioned above.

### 3.3. Curcumin (CUR)

CUR, a polyphenolic compound derived from the turmeric rhizomes of *Curcuma longa*, is commercially used as a spice and food preservative agent [111]. It also has beneficial effects on several chronic disease states linked with inflammation and oxidative stress, as observed in DM and cancer [112]. Recently, it was reported that CUR inhibits the COVID-19 protease enzyme [113]. One proposed mechanism of CUR ameliorating DM is related to its antihyperlipidemic activity via suppression of fatty-acid synthase, carnitine palmitoyltransferase 1, 3-hydroxy-3-methyl glutaryl coenzyme A reductase, and acyl-CoA cholesterol acyltransferase enzymes [114]. Moreover, CUR can diminish lipogenesis in insulin resistance syndrome, which is attributed to the inactivation of two transcription lipogenic factors: sterol regulatory element-binding protein-1-c (SREBP-1c) and carbohydrate response element-binding protein [115]. Furthermore, CUR was able to correct elevated protein-tyrosine phosphatase 1-B resulting from insulin resistance syndrome [116], leading to an improvement of the phosphorylation of insulin receptor substrate-1 (IRS-1) and JAK-2 [117], as well as suppression of STAT3 and SOCS3 [118]. CUR also stimulates Akt and ERK 1/2 [119], as well as alters the phosphatidylinositol 3-hydroxy kinase/Akt signaling pathway [120].

Moreover, the anti-inflammatory properties of CUR are attributed to its ability to inhibit macrophage infiltration and migration into metabolic organs, as well as decline some transcription inflammatory markers, including NF-κB and proinflammatory cytokines such as TNF-α, IL-1β, TLR-4, and C-reactive protein [121]. Other inflammatory indicators such as cyclooxygenase, phospholipases, and MCP-1 can be decreased in DM after the therapeutic use of CUR [122]. CUR has been found to play a role in the diabetic effect by obstructing TLR-4 activation and modifying caveolin-1 phosphorylation in diabetic patients [123].

Another effect of CUR is that it maintains mitochondrial destruction and disruption while improving mitochondrial membrane potential and biogenesis [124]. The importance of mitochondria is reflected by their role in mediating metabolic pathways and preserving cellular functions such as ion hemostasis, antioxidant defense, fatty-acid oxidation, amino-acid biosynthesis, and energy production [125]. CUR potentiates the mitochondrial activity by enhancing (i) cytochrome c protein level, which has a vital function in mitochondrial oxidative phosphorylation, and (ii) mitochondrial carnitine palmitoyltransferase 1 enzyme, which transports long-chain fatty acids into the mitochondria for β-oxidation [126].

CUR diminishes hypoxia-induced cell injury and HIF-1α, which is an oxygen-dependent conversion activator and is closely related to oxidative stress specific to diabetic cardiomyopathy [127]. CUR also plays a role in increasing wound healing in experimental diabetic rats by enhancing the expression of certain granulation tissue growth factors such as vascular endothelial growth factor (VEGF), stromal cell-derived factor-1 alpha (SDF-1α), and tumor growth factor-β1. Endothelial nitric oxide synthase was also enhanced [128]. CUR treatment was able to improve insulin sensitivity and diabetic cardiac complications via upregulation of some thermogenic genes such as uncoupling proteins 1, 2, and 3 [129], which are mitochondrial anion carriers, and it can adjust the heart’s energy metabolism and protect it against ROS by modulating mitochondrial respiration [130]. CUR treatment leads to a decrease in the accretion of S-phase kinase-associated protein 2 (S-phase Skp2) and enhances p27 protein accumulation in the pancreatic cancer cell, resulting in a significant amelioration of diabetic nephropathy [131,132]. The potential hypoglycemic role of CUR is shown in Figure 7.

### 3.4. Moringa Oleifera (MO)

MO is a persistent deciduous tropical plant belonging to the genus *Moringa* of the family Moringaceae; it is described as the marvel tree because all its parts have multiple uses in medicinal, industrial, agricultural, or functional foods [133]. The flowers, pods, leaves, and seeds of MO are regarded as food sources that contain growth promoters, as they are characterized by a high content of vitamins, minerals, and proteins [134]. From the pharmacological view, it possesses anticancer, antidiabetic, anti-inflammatory, antimicrobial, antihypertensive, and antiulcer purposes [135]. Several mechanisms contribute to the hypoglycemic curative effect of MO derived from its active constituents, particularly three classes of phytochemicals, phenolic acids (chlorogenic acid), flavonoids (quercetin and kaempferol), and glucosinolates, which have good antioxidant scavenging activity toward ROS [136]. In this regard, certain phytochemicals in MO such as quercetin and terpenoid were found to enhance glucokinase enzyme activity and pancreatic β-cells, respectively, thereby minimizing insulin resistance [137]. Due to the presence of isothiocyanates as one of its active ingredients, MO can inhibit both gluconeogenesis and glycogenolysis in the liver, as well as the absorption of glucose into adipose tissue and muscles [138]. MO also battles insulin resistance in the muscle via GLUT-4 activation, which leads to an improvement in the Akt signaling pathway [139]. On the one hand, via triggering Sirt-1, which interacts with and deacetylates peroxisome proliferator-activated receptor-1 alpha (PPAR-1α), the presence of niazirin, a phenolic glycoside in MO seeds, increases the phosphorylation of AMPKα [140]. It minimizes the levels of forkhead box protein O1 (FOXO1) and hepatocyte nuclear factor 4 alpha (HNF-4α), allowing peroxisome-proliferator activated receptor-α (PPAR-1α) to obstruct the gluconeogenesis process. Moreover, it regulates the PKC-zeta/Nox4/ROS signaling pathway that potentially decreases the oxidative stress produced in DM [141].

Furthermore, MO improves fatty-acid oxidation via the AMPK/ACC and/or PPAR-α pathways; however, it hinders triacylglycerol and cholesterol biosynthesis through sterol regulatory element-binding protein-1 (SREBP-1) regulation [142]. MO is closely related to the downregulation of α-glucosidase, pancreatic lipase, and lipoprotein lipase enzymes, which are crucial rate-restrictive enzymes obligatory for the hydrolysis of dietary carbohydrates and fats during carbohydrate and lipid metabolism [143].

### 3.5. Portulaca Oleracea (PO)

PO belongs to the Portulacaceae family. It is an annual succulent herb that grows in warm climates and is dispersed as turfgrass weed or field crop [144]. It exhibits good nutritional quality due to its high content of α-linolenic acid, ascorbic acid, β-carotene, and vitamin B complex [145]. Furthermore, it reveals a broad range of biological activities such as antiaging, antiulcerogenic, antimicrobial, antidiabetic, anticancer, anti-inflammatory, antiseptic, and neuroprotective properties, in addition to improving the immune system [146]. Here, we present several hypotheses underlying the hypoglycemic influence of PO. One such theoretical effect of PO is correlated with the promotion of insulin production in pancreatic cells via closure of potassium–ATP channels, membrane depolarization, and enhancement of Ca^2+^ influx [147]. PO also boosts glycolysis and animates phosphofructokinase, lactate dehydrogenase, and pyruvate kinase enzymes [91].

PO lessens the chronic inflammation produced due to insulin resistance through inhibition of the Rho/ROCK/NFκB pathway, which is implicated in the production of proinflammatory molecules [148]. Moreover, PO can prevent DM complications by regulating lipid metabolism via phosphorylation of ACC at Ser79, which is an AMPK phosphorylation site. As a result, fatty-acid and triacylglycerol biosynthesis is inhibited, and the PI3K/Akt and AMPK pathways in skeletal muscle are improved, resulting in increased glucose uptake in adipose tissue [149]. In addition, PO is one of the richest green plant sources of phenolic acids, flavonoids, alkaloids, triterpenoids, glutathione, and other antioxidants, making it an effective antioxidant herb for DM pancreatic cell protection [150]. As a result of its phytochemical content, especially triterpenoids and homoisoflavonoids, PO can initiate the GLUT-4 translocation [151].

### 3.6. Punica Granatum (PG)

PG is an ancient perennial plant species of the Punicaceae family, which can be found in Africa, America, Europa, and Asia [152]. The roots, barks, fruits, peels, and leaves of PG are used in numerous ailments in the treatment of cancer, microbial infections, obesity, ulcer, inflammation, and Alzheimer’s disease [153].

In general, there are several valuable PG phenolic constituents such as ellagic acid, punicalagin, flavonoids, anthocyanins, and flavonoids that provide high antioxidant capacity [154]. Polyphenols in PG play a significant role in its hypoglycemic effect via multiple pathways, including (i) improving the sensitivity of insulin receptors, (ii) increasing the activity of PPAR-γ [155] and paraoxonase 1 level, which is a high-density lipoprotein-associated lipolactonase and possesses antioxidative characters [156], (iii) modulating the expression of GLUT-4 [157], and (iv) enhancing the glucose uptake by peripheral tissues and hindering gluconeogenesis [158].

Moreover, PG inhibits the dipeptidyl peptidase-4 enzyme that is linked to glucose metabolism by degrading the incretin hormones glucagon-like peptide-1 and glucose-dependent insulinotropic polypeptide, thereby stimulating insulin secretion [159]. Furthermore, PG exhibited powerful activity in reducing glucose absorption via the inhibition of pancreatic lipase and α-amylase enzyme activities responsible for the digestion of fat and carbohydrates, respectively [160]. It was reported that PG can establish its hypoglycemic influence via inhibition of cytochrome P450 (CYP)2C9 that is responsible for the metabolization of some hypoglycemic sulfonylureas such as tolbutamide, thus increasing the efficacy of hypoglycemic drugs [161]. PG has a role in the prevention of some cardiovascular complications of DM through the suppression of lipogenesis in adipose tissue and triacylglycerol biosynthesis in the liver, as well as inhibition of fatty-acid synthase enzyme and SREBP-1c. Multiple studies have explored the antidiabetic potential of PG; one revealed the reduction in blood glucose levels and increase in insulin levels in rats by exciting β-cells and increasing their number. Another study found that, in an IDDM model treated with PG, hepatic lipid peroxidation was reduced and immune cell infiltration into pancreatic islets was inhibited [162].

The role of the abovementioned selected antidiabetic herbal plants is summarized in Table 1. Further in vivo studies of these plants are outlined in Table 2.

## 4. Conclusions

The use of medicinal plant therapy for diabetes mellitus suggests its importance in the prevention and treatment of this disease. Several herbs have displayed antidiabetic activities via various mechanisms, such as attenuating oxidative stress and inflammation, increasing insulin sensitivity and glucose uptake, and regulating insulin-induced signaling in different tissues. Furthermore, various types of herbs are readily available all over the world with low cost, low toxicity, and important phytochemical contents. Nevertheless, further clinical studies are needed to confirm the valuable effects of these plant-derived preparations in treating and managing diabetes.

## Figures and Tables

**Figure 1 molecules-26-06836-f001:**
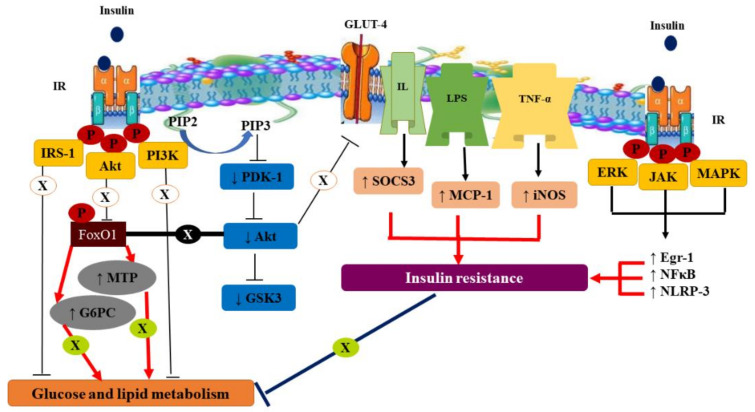
Molecular mechanism of insulin resistance. Akt: protein kinase B, IRS-1: insulin receptor substrate-1, PI3K: phosphatidyl inositol-3-kinase, PIP2: phosphatidylinositol 4,5-bisphosphate, PIP3: phosphatidylinositol 3,4,5-trisphosphate, PDK-1: phosphoinositide-dependent protein kinase 1, GSK3: glycogen synthase kinase 3, GLUT-4: glucose transporter-4, IL: interleukin, SOCS3: suppressor of the cytokine signaling, LPS: lipopolysaccharides, MCP-1: monocyte chemoattractant protein-1, TNF-α: tumor necrosis factor-alpha, iNOS: inducible nitric oxide synthase, ERK: extracellular signal-related kinase, JAK: Janus kinase-2, MAPK: mitogen-activated protein kinase, Egr-1: early growth response-1, NF-κB: nuclear factor-kappa B, NLRP-3: NOD-like receptor protein-3, FoxO1: forkhead box O1, MTP: microsomal triacylglycerol transfer protein, G6PC: glucose-6-phosphatase catalytic subunit 1.

**Figure 2 molecules-26-06836-f002:**
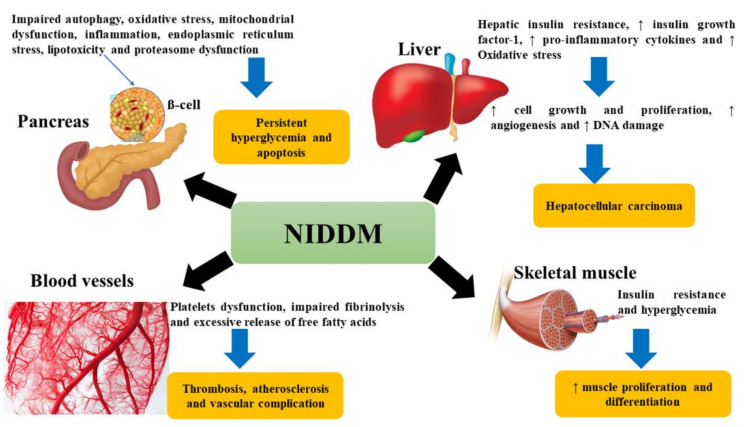
Effect of noninsulin-dependent diabetes mellitus (NIDDM) on liver, pancreas, blood vessels, and skeletal muscle. ER: endoplasmic reticulum, DNA: deoxyribonucleic acid.

**Figure 3 molecules-26-06836-f003:**
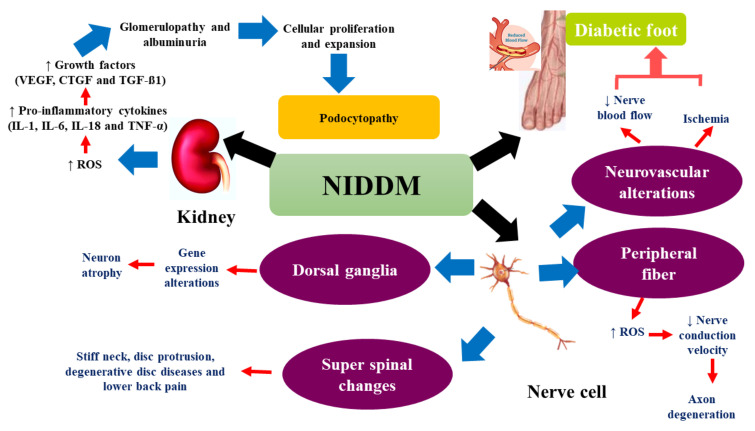
Effect of noninsulin-dependent diabetes mellitus (NIDDM) on kidney, nerve cell, and foot. VEGF: vascular endothelial growth factor, CTGF: connective tissue growth factor, TGF-β1: transforming growth factor-beta 1, IL-1: interleukin-1, IL-6: interleukin-6, IL-18: interleukin-18, TNF-α: tumor necrosis factor-alpha, ROS: reactive oxygen species.

**Figure 4 molecules-26-06836-f004:**
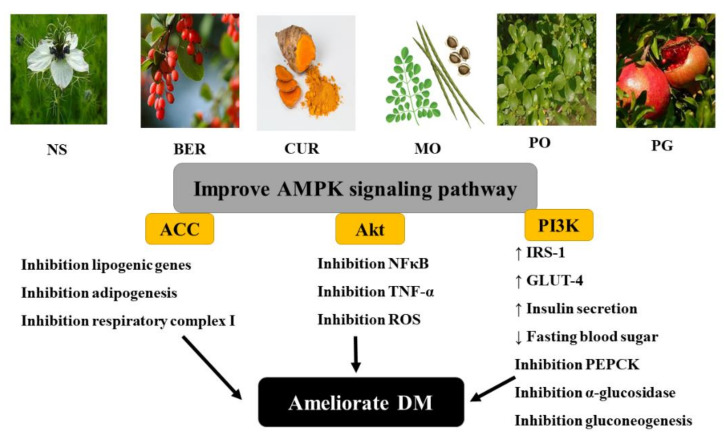
Effect of some natural plants on adenosine monophosphate-activated protein kinase (AMPK) signaling pathway. NS: *Nigella sativa*, BER: berberine, CUR: curcumin, MO: *Moringa olifera*, PO: *Portulaca oleracea*; PG: *Punica granatum*, ACC: acetyl CoA carboxylase, Akt: protein kinase B, NF-κB: nuclear factor-kappa B, TNF-α: tumor necrosis factor-alpha, ROS: reactive oxygen species, PI3K: phosphatidyl inositol-3-kinase, IRS-1: insulin receptor substrate-1, GLUT-4: glucose transporter-4, FBS: fasting blood sugar, PEPCK: phosphoenolpyruvate carboxykinase, α-glucosidase: alpha-glucosidase, DM: diabetes mellitus.

**Figure 5 molecules-26-06836-f005:**
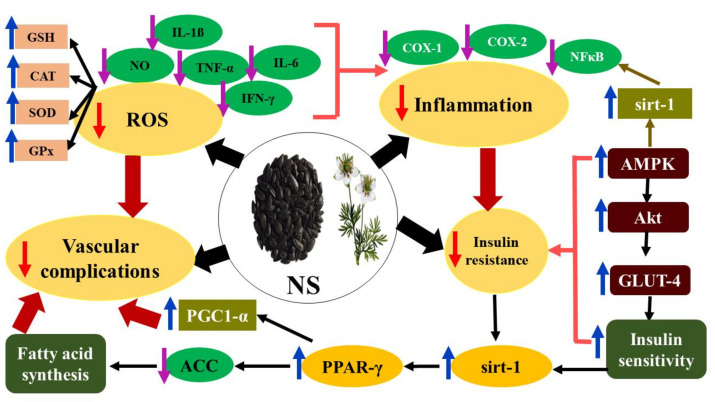
The molecular mechanistic pathways of antidiabetic effect of NS. GSH: reduced glutathione, CAT: catalase, SOD: superoxide dismutase, GPx: glutathione peroxidase, ROS: reactive oxygen species, NO: nitric oxide, IL-1β: interleukin-11 beta, TNF-α: tumor necrosis factor-alpha, IL-6: interleukin-6, IFN-γ: interferon-gamma, COX-I: cyclooxygenase-I, COX-II: cyclooxygenase-II, NF-κB: nuclear factor-kappa B, Sirt-1: Sirtuin-1, AMPK: adenosine monophosphate-activated protein kinase, Akt: protein kinase B, GLUT-4: glucose transporter-4, PPAR-γ: peroxisome proliferator-activated receptor-gamma, ACC: acetyl CoA carboxylase, PGC1-α: peroxisome proliferator-activated receptor gamma coactivator 1-alpha.

**Figure 6 molecules-26-06836-f006:**
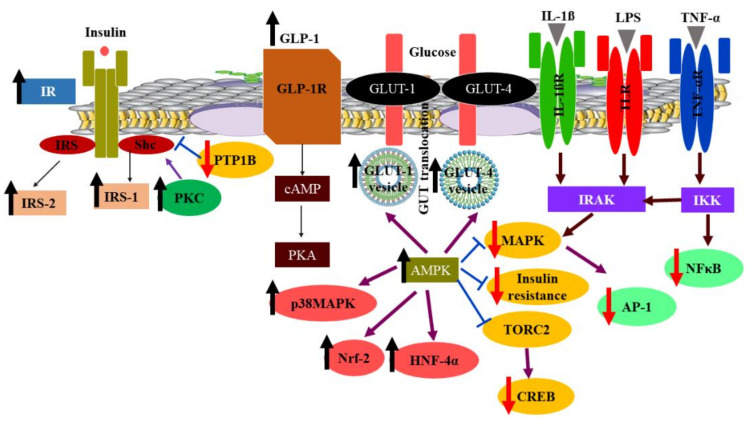
Molecular pathways of BER in ameliorating NIDDM. InsR: insulin receptor, IRS: insulin receptor substrate, IRS-1: insulin receptor substrate-1, IRS-2: insulin receptor substrate-2, Shc: mammalian Shc locus encoding three protein variants with molecular mass of 46, 52, and 66 kDa and identical modular structure, PKC: protein kinase C, PTP1B: protein tyrosine phosphatase 1B, GLP-1: glucagon-like peptide-1, GLP-IR: glucagon-like peptide-1 receptor, cAMP: cyclic adenosine monophosphate, PKA: protein kinase A, GLUT-1: glucose transporter-1, GLUT-4: glucose transporter-4, GLUT: glucose transporter, AMPK: adenosine monophosphate-activated protein kinase, p38 MAPK: p38 mitogen-activated protein kinase, Nrf2: protein regulating the expression of antioxidant proteins that protect against oxidative damage triggered by injury and inflammation, HNF-4α: hepatocyte nuclear factor-4 alpha, MAPK: mitogen-activated protein kinase, TORC2: target of rapamycin 2 kinase, CREB: cAMP response element-binding protein, IL-1β: interleukin-1 beta, IL-1βR: interleukin-1 beta receptor, LPS: lipopolysaccharides, TLR: Toll-like receptor, IRAK: interleukin-1 receptor-associated kinase, AP-1: activator protein-1, TNF-α: tumor necrosis factor-alpha, TNF-αR: tumor necrosis factor-alpha receptor, IKK: IkB kinase (a cytokine-activated protein kinase complex), NF-κB: Nuclear factor kappa B.

**Figure 7 molecules-26-06836-f007:**
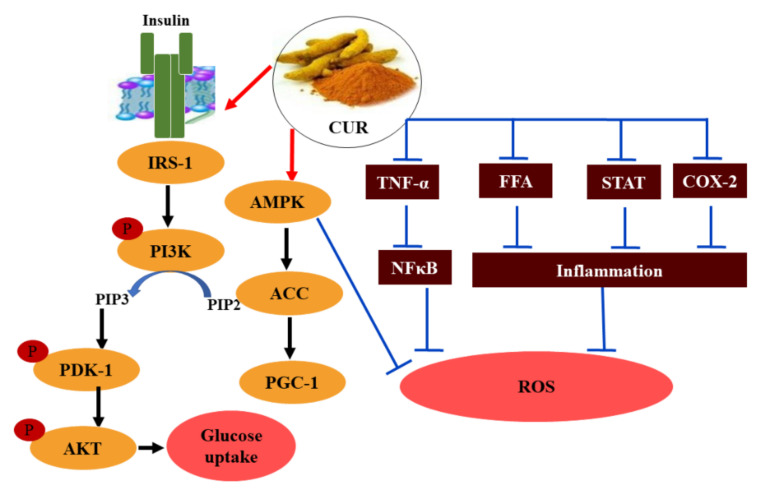
Mechanisms of the potential antidiabetic effect of CUR. IRS-1: insulin receptor substrate-1, PI3K: phosphatidyl inositol-3-kinase, PIP2: phosphatidylinositol 4,5-bisphosphate, PIP3: phosphatidylinositol 3,4,5-trisphosphate, PDKI: phosphoinositide-dependent protein kinase 1, Akt: protein kinase B, AMPK: adenosine monophosphate-activated protein kinase, ACC: acetyl CoA carboxylase, PGC-1: peroxisome proliferator-activated receptor-gamma coactivator, TNF-α: tumor necrosis factor-alpha, NF-κB: nuclear factor-kappa B, FFA: free fatty acids, STAT: Signal transducer and activator of transcription, COX-2: cyclooxygenase-2, ROS: reactive oxygen species.

**Table 1 molecules-26-06836-t001:** Antidiabetic effect of medicinal plants.

Scientific Name	Plant Family	Common Name	Traditional Use	References
*Nigella sativa*	Ranunculaceae	Black cumin	Anti-inflammatory, antidiabetic, antiparasitic, and analgesic	[163,164]
*Berberis vulgaris*	Berberidaceae	Berberine	Antihyperlipidemic, anticancer, anti-inflammatory, antioxidant, hepatoprotective, and hypoglycemic agent	[165,166]
* Curcuma longa *	Zingiberaceae	Turmeric	Anticancer, antihyperglycemic, neuroprotective, antiapoptotic, antimicrobial, and cardioprotective	[167,168]
*Moringa oleifera*	Moringaceae	Moringa	Hypoglycemic, neuroprotective, hepatoprotective, hypolipidemic, and antiviral agent	[169,170]
*Portulaca oleracea*	Portulacaceae	Purslane	Anti-inflammatory, antidiabetic, anticancer, analgesic, immunostimulant, antimicrobial, and antiviral	[171,172]

**Table 2 molecules-26-06836-t002:** Previous in vivo studies on the effect of medicinal plants on DM.

Scientific Name	Treatment Form	Dose	Fasting Blood Glucose Level (mg/dL)		Fasting Insulin Level (µIU/mL)		References
			Pre-Treatment	Post-Treatment	Pre- Treatment	Post- Treatment	
*Nigella sativa*	Oil	100 mg in 10% DMSO/kg Bwt *	581.31 ± 36.31	142.76 ± 16.94	101.59 ± 5.78	127.86 ± 1.27	[73]
*Berberis vulgaris*	Berberine chloride	100 mg/kg Bwt	180.1 ± 4.38	97.7 ± 5.61	20.17 ± 2.93	15.67 ± 2.42	[97]
*Curcuma longa*	Curcumin	50 mg/kg Bwt	481 ± 0.71	109.20 ± 0.86	180.44 ± 0.43	80.44 ± 0.15	[173]
*Moringa oleifera*	Ethanolic extract of leaves	200 mg/kg Bwt	167.0 ± 1.96	94.0 ± 4.96	45.03 ± 13.8	36.4 ± 4.66	[174]
*Portulaca oleracea*	Water extract	250 mg/kg Bwt	293.2 ± 2.4	125.0 ± 1.3	18.97 ± 0.09	33.50 ± 0.08	[175]

* Bwt: body weight.

## Data Availability

Data sharing is not applicable.

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
