# Peer review of "Therapeutic Screening of Herbal Remedies for the Management of Diabetes"

_molecules, 2021, doi:10.3390/molecules26226836_

Round 1
Reviewer 1 Report
In this study, Mahmoud Balbaa1 and co-workers summarize the current prevalence of diabetic mellitus (DM), pathophysiology, and molecular mechanism, and diabetic complications, and the side effects of conventional treatment for DM. This work focuses on herbal medicines as the new therapy for the management of DM, herbal plants include Nigella sativa, berberine, curcumin, moringa olifera, moringa olifera, and Punica granatum as alternative therapy for DM. In general, the review systematically describes and provides a large number of statements to support the conclusions. However, there are a several issues that still need to be addressed.
- it is reported that multiple factors such as oxidative stress, inflammation, and hyperglycemia are implanted in the pathogeneses of IR syndrome. Although Figure 1 illustrated part molecular mechanisms such as GLUT-4, NLRP3, and AKT/PI3K signaling pathways in IR syndrome, it is not summarized comprehensively. Therefore, it is necessary to add other pathways (e.g. FoxO1-mediated hyperglycemia) as well as the correlation between pathways.
- The development of DM can result in long-term complications as evidenced by the damaged organs. Figure 2 showed the effect of NIDDM on organs such as liver, pancreas, and skeletal muscle. other diabetic complications such as diabetic nephropathy, diabetic neuropathy and diabetic foot also should be introduced in Figure 2.
- In this review, AMPK has been shown to paly the central regulator in the mechanism of herbal plants improve hyperglycemia and IR. But there is little content about AMPK and its mediated signaling pathway. Thus, detail introduction of the role of APMK on DM and its related pathway is necessary.
- Tables are recommended to summarize the anti-diabetic effect of medicinal plants (Nigella sativa, berberine, curcumin, moringa olifera, moringa olifera, and Punica granatum), including the plant's scientific names, family, common name, traditional use, and references.
Author Response
- It is reported that multiple factors such as oxidative stress, inflammation, and hyperglycemia are implanted in the pathogeneses of IR syndrome. Although Figure 1 illustrated part molecular mechanisms such as GLUT-4, NLRP3, and AKT/PI3K signaling pathways in IR syndrome, it is not summarized comprehensively. Therefore, it is necessary to add other pathways (g. FoxO1-mediated hyperglycemia) as well as the correlation between pathways.
Thanks, for your suggestion, we add the FoxO1-mediated hyperglycemia pathway in Figure 1 and also the correlation between different pathways performed. All changes are yellow highlighted in the manuscript.
- The development of DM can result in long-term complications as evidenced by the damaged organs. Figure 2 showed the effect of NIDDM on organs such as the liver, pancreas, and skeletal muscle. other diabetic complications such as diabetic nephropathy, diabetic neuropathy and diabetic foot also should be introduced in Figure 2.
Thanks for your suggestion. Diabetic nephropathy, diabetic neuropathy, and diabetic foot had been illustrated in the new Figure (Figure 3) in the revised manuscript. All changes are yellow highlighted in the manuscript.
- In this review, AMPK has been shown to play the central regulator in the mechanism of herbal plants to improve hyperglycemia and IR. But there is little content about AMPK and its mediated signaling pathway. Thus, a detailed introduction of the role of APMK on DM and its related pathway is necessary.
Thanks for your comment. The role of AMPK on DM is fully illustrated in the manuscript. Also, the pathways in Figures 5, 6 & 7 show the role of AMPK. All changes are yellow highlighted in the manuscript.
- Tables are recommended to summarize the anti-diabetic effect of medicinal plants (Nigella sativa, berberine, curcumin, Moringa olifera, and Punica granatum), including the plant's scientific names, family, common name, traditional use, and references.
Thanks for your advice. The antidiabetic effect of medicinal plants is summarized in table 1 in the manuscript. All changes are yellow highlighted in the manuscript.
Reviewer 2 Report
Journal: Molecules
Manuscript ID: molecules-1454315
Type of manuscript: Review
Title: Therapeutic screening of herbal remedies for the management of diabetes
Authors: Mahmoud Balbaa, Marwa El-Zeftawy, Shaymaa A Abdulmalek
In the present review article, recent progress on the effectiveness and protection of natural management of diabetes mellitus (DM) have been discussed, with the common herbs used in the prevention and treatment of DM presented. This manuscript provides some important information for the potential development of anti-diabetes agents from plants, and thus it is recommended to be published as a Review in Molecules after minor revision shown below and in the attached manuscript pdf file.
- The authors need to include in vivo investigations for the herbal products selected in the manuscript if they were reported.
- Page 3 line 87: Update “NF-kB” as “NF-κB” over the manuscript.
- Page 14 line 432: Remove the issue number and change the volume number to be italic for all references.
- English should be improved.
- Revise the article following the attached pdf manuscript file.

Author Response
- The authors need to include in vivo investigations for the herbal products selected in the manuscript if they were reported.
Thanks for your advice. The previous in vivo investigations of mentioned medicinal plants are added in table 2 in the manuscript. All changes are yellow highlighted in the manuscript. In fact, many antidiabetic effects were done in the animal model of experimental diabetes that is a real example of in vivo study.
- Page 3 line 87: Update “NF-kB” as “NF-κB” over the manuscript.
Thanks for your comment. NF-κB was updated in all pathways and the whole manuscripts.
- Page 14 line 432: Remove the issue number and change the volume number to be italic for all references.
Thanks for your comment. All issues in the reference section were removed and all volume numbers changed to italic and yellow highlighted.
- English should be improved
Thanks for your recommendation. The whole manuscript was checked for sentence structure and grammar rules. Many sentences were corrected or rephrased to improve the meanings (yellow-labelled).